# Tensile Behavior of Chain Links Made of Polymeric Materials Manufactured by 3D Printing

**DOI:** 10.3390/polym15153178

**Published:** 2023-07-26

**Authors:** Bruno Rădulescu, Andrei Marius Mihalache, Emilian Păduraru, Adelina Hriţuc, Mara Cristina Rădulescu, Laurenţiu Slătineanu, Vasile Ermolai

**Affiliations:** 1Department of Digital Production Systems, “Gheorghe Asachi” Technical University of Iași, 700050 Iași, Romania; bruno.radulescu@academic.tuiasi.ro (B.R.); emilian.paduraru@academic.tuiasi.ro (E.P.); mara.radulescu@academic.tuiasi.ro (M.C.R.); 2Department of Machine Manufacturing Technology, “Gheorghe Asachi” Technical University of Iași, 700050 Iași, Romania; marius-andrei.mihalache@academic.tuiasi.ro (A.M.M.); slati@tcm.tuiasi.ro (L.S.); vasile.ermolai@student.tuiasi.ro (V.E.)

**Keywords:** chain link, polymers, tensile testing, influence factors, empirical mathematical models

## Abstract

For reduced mechanical stress, some chains with links made of metallic materials could be replaced by chains made of polymeric materials. A lower weight and a higher corrosion resistance would characterize such chains. From this point of view, research on the behavior of chain links made of polymeric materials under the action of tensile stresses can become important. Modeling by the finite element method highlighted some specific aspects of the behavior of a chain link subjected to tensile stresses. Later, we resorted to the manufacture by 3D printing of some chain links from four distinct polymeric materials, with the modification of the size of the chain link and, respectively, of the values of some of the input factors in the 3D printing process. The tensile strength of the chain links was determined using specialized equipment. The experimental results were processed mathematically to determine some empirical mathematical models that highlight the influence of the values of the input factors in the 3D printing process on the tensile strength of the samples in the form of chain links. It thus became possible to compare the results obtained for the four polymeric materials considered and identify the polymeric material that provides the highest tensile strength of the sample in the form of a chain link. The results of the experimental research showed that the highest mechanical resistance was obtained in the case of the links made of polyethylene terephthalate glycol (PETG). According to experimental results, when tested under identical conditions, PETG links can break for a force value of 40.9 N. In comparison, polylactic acid links will break for a force value of 4.70 N. Links printed in the horizontal position were almost 9-fold stronger than those printed in the vertical position. Under the same test conditions, according to the determined empirical mathematical models, PETG links printed in a horizontal position will break for a force of 300.8 N, while links printed in a vertical position will break for force values of 35.8 N.

## 1. Introduction

The chain link is a component of the chain. In turn, the chain is a string of links assembled consecutively between them and which is used to materialize some tying, suspension, tensile, etc., operations. The previous definition of a chain also highlights its main possibilities of use or the stress it is subjected to.

In its simplest form, a chain link is shaped similar to a ring deformed to reduce the width of the chain link and, thus, of the chain. The material and dimensions of a chain link in a cross-section are usually determined considering the mechanical stresses to which the chain composed of links will be subjected.

The classification of chain links can be carried out taking into account the shape of the chain links (circular or oval), the destination of the chain (for limiting the movements of an object or animal, for suspending or towing an object or an assembly/subassembly, for increasing the towing capacity of tires (in the case of anti-skid chains)), etc.

Given the chains’ relatively high tensile stress values, their links are made of metallic materials, mainly steel. The need to reduce the chains’ weight, especially when the mechanical stresses are somewhat lower, has led to the research and use of chain links made of plastic or composite materials.

Plastic materials are non-metallic materials with an amorphous structure obtained by melting together several constituents, such as resins, plasticizers, dyes, lubricants, fillers, or auxiliary materials. These materials are characterized by the possibility of being easily shaped, thanks to good plasticity, at temperatures usually between 140 and 180 °C. On the other hand, composite materials include a metallic or non-metallic mass and are reinforced with resistance elements from the category of short or very short fibers, long fibers, fabrics, felt, etc.

Plastic or polymer matrix composite chain links are lighter than metal chain links and have higher corrosion resistance. Consistent with the general properties of plastics, such materials are thermally and electrically insulating [1,2].

Suppose the production of chain links from metallic materials is based on bending a workpiece originally as a bar or wire. In that case, the chain links from plastic or composite materials can be obtained through specific processes, such as injection and 3D printing.

The expansion in the last decades of 3D printing processes has facilitated the development of some research on the properties required by the uses of chains with chain links made of plastic materials or composite materials with polymer matrix made even by 3D printing.

Thus, the possibilities of direct manufacturing through 3D printing of interconnected moving parts in the category of which interconnected chain links are also highlighted by researchers [3,4,5].

Chains with links having more complicated shapes were made by 3D printing (SLM) and studied by Venes et al. [6]. The material of the chain links was an alloy containing titanium, aluminum, and vanadium. It was found that it is possible to ensure a clearance between two moving surfaces of 150 µm.

Wójcicki et al. designed and built a stand for the wear testing of the chain links of the scraper conveyor [7]. A simplified stand prototype was materialized using 3D printing of its components.

The possibility of making chain links from lightweight material through additive manufacturing was highlighted [5,8]. Chains of this type can be used in the mining industry to reduce the weight of some equipment.

A theoretical and experimental analysis of the stress generated in metal chain links under different operating conditions was carried out by Mešić et al. [9]. They considered stress analysis for different chain link positions during mechanical loading.

The possibilities of manufacturing some objects from ceramic materials with polymeric matrix and including some chain links from such materials were investigated by Román-Manso et al. They resorted to 3D printing with microwave-activated polymerization [10].

3D printing of polymer link chains for dog harnesses was addressed by Woodman [11]. He produced a report outlining the possibilities of using 3D printing to create objects that could improve the living conditions of pets or laboratory animals. He thus analyzed the situation of the use of chain links made of plastic materials, appreciating that the plastic material of the links must be hard enough so that the animals do not swallow the particles detached from the links as a result of use and harm their health.

The finite element method was used by Noguchi et al. for a static stress study in the case of a link plate of a roller chain [12]. The study also aimed at identifying some possibilities for reducing the weight of the chain.

Researching the behavior of ship mooring chains on shore has been an important objective for different groups of researchers [13,14,15,16,17,18,19,20,21,22,23,24,25,26]. The static, dynamic, and fatigue stresses occurring in the chain links for different operating conditions were thus highlighted.

The values of some input factors in the additive manufacturing process are expected to influence the physical-mechanical properties of the materials of the parts manufactured by such processes [8,9,27].

The problem of manufacturing chain links from polymeric materials was an objective also addressed in various invention patents [28,29,30].

From those mentioned above, it can be seen that until now, the research related to the chain links made of metallic and non-metallic materials has focused on the establishment of the dimensions of these chain links [1,7,31,32,33], the efforts generated in the chain links during operation [8,18,25,26,31], the modeling by the finite element method or by other methods of the stresses and deformations from chain links during their mechanical loading [9,12,13,14,15,18,19], investigating the fatigue behavior of chain links [20,21,22].

It was found that there is relatively little information on the behavior of chain links from polymeric materials manufactured by 3D printing when these chain links are subjected to tensile stress. It is necessary to mention that in the category of chains made of metallic materials that could be replaced by links made of polymers or chains with links made of polymeric materials, some chains for ornaments of monuments, chains for suspending some objects of religious worship, chains of ornaments, chains for leashes used to tie dogs, etc.

The authors of this paper are unaware of results obtained by other researchers regarding the influence exerted by the dimensions of polymer material beads and the factors that characterize the manufacturing conditions of beads by 3D printing on the tensile strength of beads. For low mechanical loads, link chains made of polymeric materials could be characterized by lower mass and superior corrosion resistance. Such chains could be made directly by 3D printing without the need for an assembly operation of separately manufactured chain links.

The finite element method was used to obtain additional information regarding the behavior of the chain links during their tensile testing. In the framework of this paper, the main aspects related to oval-shaped rings manufactured by 3D printing processes from polymeric materials will be briefly analyzed. The conditions for conducting the tensile tests of the 3D printed polymer materials and the obtained experimental results will be presented in the next chapter. Afterward, the emphasis will be on the mathematical processing of the experimental results and the identification, as such, of some empirical mathematical models that characterize the tensile behavior of chain links made of polymeric materials and, respectively, of a composite material with a polymeric matrix. The analysis of the empirical mathematical models will have to allow the formulation of some observations regarding the meaning of variation and the intensity of the influence exerted on the maximum force supported by the chain links when the sizes of the input factors in the investigated process change.

## 2. Materials and Methods

### 2.1. The Mechanical Loading of a Chain Link

As mentioned, the chain link is shaped similar to an ovalized ring, with a cross-section through the ring usually revealing a circular shape. During operation, static or dynamic stretching, compression, bending, and torsional stresses can occur in different areas of the chain link.

From the point of view of most uses of the chain link, its tensile strength is of particular interest. However, even during a tensile test, different areas of the chain link may be affected by other categories of stress.

In principle, the tensile testing of the chain link involves using hooks or bolts whose ends are inserted into the gap of the chain link and through which tensile stress is generated in the lateral areas (Figure 1).

The mentioned stress can have a static character. Compressive and crushing stress occurs in the contact areas of the chain link with the two bolts. Bending efforts occur near the contact between the chain link and the bolts through which the chain links are subjected to mechanical stresses. Torsional stresses will still appear in chain links if a twisted chain is considered. Still, there are situations when the loading can also be cyclical, which requires some research on the fatigue resistance of the chain links.

The manufacture of chains from oval-shaped metal chain links may involve operations of cutting straight segments from wire or bar, heating, and descaling, and one or more bending operations, with the eventual joining of chain links, butt welding, shot blasting, marking, thermal treatment, application of surface protection operations, etc. [26]. Chain manufacturing technology also includes some operations to inspect and test the quality of products in different phases of the manufacturing process. Stretch testing and impact testing are such chain quality control operations.

Chain links can be made of polymer or matrix composite materials for low or moderate loads. Thus, chain links can be manufactured by injection molding. As mentioned, the expansion in the last decades of 3D printing processes led, among other results, to ensure the conditions for manufacturing chain links from polymer materials or composite materials with a polymer matrix through additive manufacturing processes. The possibility of changing the values of some parameters specific to the manufacturing conditions of the chain links through 3D printing processes facilitates a deeper investigation of the factors capable of influencing the behavior of the chain link during their operation.

Fused deposition modeling is one of the most widespread groups of techniques that can be used to manufacture parts from polymer materials through 3D printing. In this case, the material of a polymer wire is melted and then deposited to make up the successive layers of material of a particular part. Let us note that fused deposition modeling or 3D printing, in general, can provide conditions for manufacturing the assembled chain links from the beginning so that the chain can be directly obtained.

An improvement in the behavior of chains made of non-metallic materials becomes possible through composite materials with a polymer matrix, in which case the use of certain reinforcement materials could contribute to a significant improvement in the behavior of the chains during their exploitation.

Suppose the manufacture of chain links by fused deposition modeling is considered. In that case, the main groups of factors capable of affecting the behavior of the chain links or chains during their use are the following:
The sizes and shapes of the chain links;The nature and physical-mechanical properties of the polymer material or the composite material with a polymer matrix;Some parameters that characterize the conditions for carrying out the 3D printing process;The way to load chain links, the nature of these loads, etc.


It is interesting to study the extent to which some of the previously mentioned factors or groups of factors influence the mechanical strength characteristics of chain links made of polymer materials or polymer matrix composite materials.

### 2.2. Finite Element Modeling of the Behavior of a Plastic or Composite Chain Link

A chain link with the shape and dimensions indicated in Figure 2 was considered. One criterion considered when determining the geometry and dimensions of the 3D printed and tensile-tested chain links was to approximate the dimensions and geometry of existing chain links in practice. Since it was also intended to highlight the influence exerted by the size of the diameter of the za element on the tensile behavior, we resorted to the use of two such diameters, with values close to those existing in the case of some chains encountered in practice.

As a matter of choice, the finite element method (FEM) is to obtain a similar crack pattern as one from the experimental tests. This would give insight into stress distribution just before the crack extends on the whole part length and the fracture occurs. For this purpose, it had to design a similar model with one of the eight test samples used in the experimental tests. It had chosen sample 1 with a 30% infill (see Figure 3a). This option is because sample 1 is printed vertically (see Table 2), making it more difficult to print and analyze than its horizontal peers. The E-moduli obtained using the fused deposition modeling (FDM) printing process is lower than that provided by the manufacturer and depends on the printing direction. Six samples of Ultrafuse ABS-type filament manufactured by BASF (Ludwigshafen, Germany) on Instron 4411 equipment were printed for analyses to be as accurate as possible and tested. It carries a 5 kN load cell with a 5 mm/sec testing speed. The room temperature was 22 °C with a 60% humidity level. It resulted in an E-modulus mean value of 558 MPa. The manufacturer lists in the filament’s technical data sheet (TDS) 1608 MPa on ZX direction of printing [34], whereas the Ansys library uses 1628 MPa. The measured value was used inside Engineering Data, Ansys’s library of materials. The infill model was replicated in a 3D model and later translated into a Parasolid file for analysis (see Figure 3b).

The 3D model also received a crack initiation model because the FEM analyses are based on the prediction of crack propagation using Ansys’s SMART Crack Growth, which relies on the energy release rate method. It has opted for the stress intensity factor as the desired fracture criterion inside a static crack growth mechanism. This one predicts a pre-existing crack growth when certain loading conditions are met. The algorithm calculates the maximum stress intensity factor of the crack’s front nodes, and if it exceeds the specified criterion, the crack is set to grow. The critical rate was set to 100 MPa mm^1/2^, which will not stop on maximum crack extension. The body was assigned an ABS type of material retrieved from the software’s library. It has imposed a virtual topology that uses the edges-only type of behavior. Mesh controls use a patch-conforming method with quadratic element order set for the entire chain link body and a face sizing of 0.15 mm for the two faces of the pre-existing designed crack. Element size was set to 0.25 mm for the entire mesh, with an aggressive mechanical method set for whenever error limits are encountered during analyses. That gave almost 80.000 elements and more than 140,000 nodes (see Figure 4a). The pre-meshed crack was designed to occur before the infill pattern, thus giving us the opportunity to set the front, top and bottom nodes using named selection. As experimental tests revealed in the case of sample 1 the crack initiates at the beginning of the semi-circular top section of the chain link. After several trial and error attempts, it has chosen 6 solution contours and imposed a new Coordinate System set for the crack only for the algorithm to work properly. This system has the X axis set in the direction of the crack extension as the Y sits almost normal on its geometry (see Figure 4b). Conditions use a fixed support for the lower section of the chain link and a force value imposed on the upper section of the chain link in the Z+ direction of the Global Coordinate System (see Figure 4c). Analyses settings are set to 0.00001 sec end step time divided into ten sub-steps.

Just before separation, the value of force required by the software to generate crack growth was 54 N which is close to that registered in the case of sample 1 made of ABS. (Table 2). The difference may come from the fact that the 3D model uses an ideal type of homogeneity, whereas the printed sample suffers from this perspective. It has resulted in 15.025 MPa registered for the equivalent stress evaluated using Von-Mises criteria. The graphical representation shows a distribution along the fracture line with its ends colored in red (see Figure 5a). Equivalent elastic strain peaks at 0.042454 mm/mm, showing that the chain link takes the force solicitation and tries to overcome it by distributing it along its length (see Figure 5b). Finally, it has attached a real-life photo of cracked sample 1 after the experimental test. It can observe a similar direction of propagation as that obtained for the pre-designed crack (Figure 5c).

SMART crack growth produces a result similar to that obtained using experimental tests. Before the body gets separated, stress and strain distribution is very useful information for future research, which would consider optimization using 3D printing setup and design. However, results should be used carefully. Further refinement is necessary, considering a finer mesh, more contour plots, or another type of crack growth assessment with an arbitrary or surface-based crack.

### 2.3. Experimental Conditions

The objective pursued through the experimental research was to highlight the influence exerted by the nature of some polymer materials used for manufacturing chain links, some dimensions of the chain links, and some factors that characterize the conditions of 3D printing of chain links.

Four materials were considered, namely: acrylonitrile butadiene styrene (ABS, Kimya, Nantes, France), acrylonitrile butadiene styrene-Kevlar composite material (ABS-K), polylactic acid (PLA), and polyethylene terephthalate glycol (PETG) (Prusa, Prague, Czech Republic). Information on some of the properties of these materials has been listed in Table 1.

The chain links were manufactured by 3D printing on a Prusa i3MK3S equipment made in the Czech Republic. The values of some input factors in the 3D printing process have been established, as shown next. The values of the printing speed *v*, the thickness *t* of the deposited layer, and those of the infill density *i* were entered into the 3D printing program. As far as possible, consideration was given to the use of the same input factor values for the experiments performed on each of the four materials from which the chain links test samples were manufactured. Instead, for the printer plate temperature *θ_p_* and the extrusion temperature *θ_e_*, a rewrite of the 3D printing program was required.

The equipment used to determine the mechanical strength of the beads was a tensile testing machine type LRX Plus (Lloyd Instrument Ltd., An AMETEK Company, Hampshire, UK). It uses XLC-5000-Al type of load cell up to 5 kN with a precision of 0.5% according to ASTM E4 and DIN 1221. Data acquisition was made by using equipment’s NEXIGEN Data Analysis software (ver. 3.0) solution.

The components involved in the experimental research can be seen in Figure 6.
polymers-15-03178-t002_Table 2Table 2Conditions for performing experimental tests and results of tensile tests.Part/ Exp no.Values of the Input FactorsValues of the Output Parameter (Maximum Force *F*, N)Chain Link Diameter, *d*, mmPrinting Speed, *v*, mm/sLayer Thickness, *t*, mmInfill Density, *i*, %Temperature Build Plate, *θ_p_*, °CExtrusion Temperature, *θ_e_*, °CPrinting Position, *p*ABSABS-KPLAPETG13800.23090245153.80951.4154.700540.9623800.2601102652350.23216.41235.7414.93331200.330902652459.4594.09252.02414.3431200.360110245154.51947.876302.216.92354800.3301102452435.48208.47403.73273.5664800.360902651120.03124.44121.7880.394741200.2301102651150.47113.51141.96103.43841200.260902452289.04233.66257.23612.66Printing position: *p* = 1 for vertical position; *p* = 2 for horizontal position.


Clamping of the bars to the tensile testing machine was carried out using 2 bolts (Figure 7).

The deformation rate of the specimens during the tensile test was 6 mm/min. The LRX Plus Universal Tensile Testing Machine can obtain the force–deformation diagram and all related numerical information directly.

To obtain as complete information as possible regarding the factors that influence the tensile behavior of the chain links manufactured by 3D printing, they resorted to using a Taguchi fractional factorial experiment of type L8 (2^8–1^), with 7 input factors at two variation levels. Each type of material received an additional sample for testing purposes to proof our L8-1 plan. Such an experimental program also ensures a certain reduction in the number of experimental tests necessary to be performed without the precision of the established empirical mathematical model being significantly affected [35,36].

The 7 input factors considered were the chain link diameter (*d_min_* = 3 mm, *d_max_* = 4 mm), printing speed (*v_min_* = 80 mm/s, *v_max_* = 120 mm/s), layer thickness (*t_min_* = 0.2 mm, *t_max_* = 0.3 mm), infill density (*i_min_* = 30%, *i_max_* = 60%), build plate temperature (*θ_pmin_* = 90°, *θ*_pmax_ = 110°), extrusion temperature (*θ_emin_* = 25 °C, *θ_emax_* = 265 °C and, respectively, the way of specimen placement during the 3D printing process (code 1 for vertical printing and code 2 for horizontal printing).

Experimental research has shown that the duration of the printing process of a vertically placed specimen is approximately 50% longer than that corresponding to printing the specimen in a horizontal position. An image made as a screenshot and related to the vertical or horizontal placement of the chain link in the 3D printing process is shown in Figure 8. In this image, the support elements generated by the slicer were highlighted using green color.

The value of the maximum force *F* at which the initiation of the sample damage process is highlighted was used as the output parameter. The values of the *F* forces for each material and experimental test were determined by analyzing the force–deformation diagrams generated by the computer program used by the tensile testing machine.

In Figure 9, the aspect of a chain link can be observed before being subjected to the tensile test and, respectively, after stopping the tensile test.

In Figure 10, the 32 specimens manufactured by 3D printing from different polymer materials can be observed after their rupture by the tensile test.

Some examples of force–deformation curves made under the conditions of testing the specimen made of ABS polymer material can be seen in Figure 11.

The influence of the diameter d on the magnitude of the maximum force F that determines the damage of a chain link made of different materials can be seen in Figure 12.

The conditions for performing the experimental tests were those mentioned in Table 2. Each experimental trial was performed only once, so the experimental trials were not repeated with the same values of the input factors in the 3D printing process. It is mentioned that the design of the experiment method (used to design the experimental trials whose experimental results were included in Table 2) was introduced and used to allow obtaining the maximum information with a minimum of trials [37,38,39,40].

The values of the force *F* determined by analyzing the diagrams corresponding to each chain link made of the 4 materials considered were entered in the last 4 columns of Table 2.

The experimental results were processed using a computer program based on the least squares method [41]. The computer program can select the most appropriate empirical mathematical model from among five models (first-degree polynomial, second-degree polynomial, power-type function, exponential-type function, hyperbolic-type function).

The selection is made by using Gauss’s criterion value [41,42]. This value of Gauss’s criterion is determined by taking into account, first of all, the sum of the squares of the differences between the values of the ordinates corresponding to the use of the proposed mathematical model and, respectively, the values of the ordinates related to the experimental results, for the same values of the abscissas. The sum of the squares of the mentioned differences is related to the difference between the number of experimental trials and the number of constants in the dependence relationship [41,42]. A lower value of Gauss’s criterion characterizes a better approximation of the experimental results to those calculated using a certain empirical mathematical model.

Among the 5 empirical mathematical models that can be determined using the mentioned computer program, the power-type function was preferred, given its use for the characterization of other processes or sizes specific to machine manufacturing (for example, for determining the tool life, cutting force components, and roughness parameters of the machined surfaces). It is necessary to note that power-type functions are preferred when it is considered that there is a monotonous variation in the pursued output parameter to the variation in the input factors in the investigated process. It will be assumed, as such, that in the situations presented in this paper and for the variation intervals considered, the seven input factors do not determine the occurrence of maxima or minima of the value of the force *F*, and they lead to a monotonous variation in the output parameter. The selection of a factorial experiment of type L8-2^7^ to arrive at empirical mathematical models that highlight the influence of different factors on the force *F* was carried out, taking into account the simplification of the calculations necessary to identify empirical mathematical models [43,44,45,46].

A monotonic variation, at least for certain ranges of variation in the values of the input factors in the 3D printing process, was highlighted by the experimental results obtained by other researchers. Thus, Vanaei et al. considered that changes in the crystallization mode could explain the variation in ultimate strength of the liquefier temperature and, therefore, the material’s microstructure in the sample [47]. For a temperature variation between 200 and 220 °C, they found an increase in ultimate strength in the case of samples made of polylactic acid. Similar explanations were formulated by Vanaei et al. regarding the influence exerted on ultimate strength by print speed, support temperature, and layer height. The influence of different input factors in the 3D printing process on ultimate strength was investigated by Jackson et al. [48]. They thus considered that increasing the ultimate strength by modifying the deposition angle could be connected with the retraction speed. The 3D printed parts with higher retraction speed presented higher values of the ultimate strength, which means a monotonous variation in the ultimate strength when the retraction speed is increased. Meram and Sözen appreciated that lower layer thickness values lead to higher ultimate strength values, ensuring better adhesion between layers [49]. This means that the two researchers thought there was a monotonous variation in the ultimate strength when changing the value of the thickness of the deposited material layer. A contrary result was obtained by Cho et al., which shows that the higher the thickness of the deposited layer, the higher the mechanical strength of the deposited material [50]. They, therefore, accept a monotonous variation in the mechanical resistance of the material of the specimen made by 3D printing by the thickness of the deposited layer. Müller et al. observed a monotonic variation (an increase) in tensile strength when the infill density increases [51].

An advantage of using empirical mathematical power-type function models is that such models provide direct information on the intensity of influence exerted by an input factor by comparing the value of the exponent attached to that factor in the mathematical power-type function with the values of other exponents. At the same time, a positive value of the exponent in question means that for the range of variation investigated, an increase in the size of the input factor will lead to an increase in the value of the output parameter. In contrast, a negative value of the exponent will cause a decreasing the size of the output parameter.

The mathematical relationships corresponding to the force *F* and the values of Gauss’s coefficients are mentioned next.

Thus, in the case of ABS material, the mathematical relationship was determined: the value of Gauss’ criterion being *S_G_* = 0.115262.
*F* = 9.436∙10^−16^*d*^1.366^*v*^0.0622^*t*^0.288^_i_^−0.322^*θ_p_*^0.471^*θ_e_*^6.588^*p*^2.144^(1)

The empirical mathematical model determined for the ABS-K material has the form:*F* = 5.166∙10^−8^*d*^2.276^*v*^−0.543^*t*^−0.571^*i*^0.349^*θ_p_*^0.692^*θ_e_*^2.795^*p*^1.208^(2)
in this case, Gauss’s criterion being *S_G_* = 8.626843∙10^−3^.

For the PLA material, the following form of the empirical mathematical model was arrived at:*F* = 3.973∙10^−32^*d*^2.656^*v*^2.424^t^2.791^*i*^1.259^*θ_p_*^5.856^*θ_e_*^6.209^*p*^1.993^(3)
the Gauss criterion having, in this case, the value *S_G_* = 0.28817645.

For the PETG material, the empirical mathematical model is of the form:*F* = 1.158∙10^−16^*d*^2.137^*v*^0.106^*t*^−1.118^*i*^−0.118^*θ_p_*^−1.787^*θ_e_*^8.017^*p*^3.071^(4)
the Gauss criterion having, in this case, the value *S_G_* = 0.2780781.

## 3. Discussion

The graphical representations in Figure 12 and Figure 13 were developed using empirical mathematical models.

The analysis of the experimental results, the empirical mathematical models, and the graphic representations elaborated on their basis allowed the formulation of the observations mentioned below.

Examining the empirical mathematical models (Equations (1)–(4)) reveals maintenance of the same direction of force magnitude variation only when considering the corresponding diameter *d* of a cross-section through the chain link and, respectively, the position in which the chain link was manufactured by 3D printing on the printer table.

As expected, an increase in the diameter *d* of the cross-section through the chain link increases the maximum force *F* (Figure 11). Still, following the initial hypothesis, manufacturing the chain link in a horizontal position leads to a higher strength of the chain link and a higher value of the maximum force *F*. This fact can be explained by the better behavior of the applied material layers to the stretching stress right along the main stress direction on the chain link during the tensile test. The continuity of the layers formed by the gradual deposition of the wire of melted polymeric material leads to an increase in the tensile strength of the polymer material.

Less expected results were obtained in the case of chain links made of ABS material with Kevlar when a decrease in the mechanical resistance of the chain links reinforced with one of the most resistant plastic materials (Kevlar) was observed. The analysis of the appearance of the broken test samples as a result of tensile stress reveals a certain separation of the ABS polymer from the Kevlar fibers due to a reduced adhesion between the two materials. Since the Kevlar fibers are not continuous, it is possible that the low adhesion between the materials incorporated in the chain link and the discontinuity of the Kevlar fiber led to the decrease in the mechanical strength of the ABS-K material chain links, compared to the mechanical strength of the chain links made of ABS material only. It was also observed that under the conditions considered for the 3D printing process, the ABS-K type material became brittle, which led to the destruction of the tensile chain link at the appearance of the first cracks. The lower mechanical strength of the chain links made from PLA may be due to the 3D printing conditions. The results of the experimental research showed that the highest mechanical resistance was obtained in the case of the links made of polyethylene terephthalate glycol (PETG). According to the experimental results, when tested under identical experimental conditions (experiment no. 1 in Table 2), PETG links can break for a force value of 40.9 N. In comparison, polylactic acid links will break for a force value of 4.70 N. Links printed in the horizontal position were almost 9-fold stronger than those printed in the vertical position. Under the same test conditions (*d* = 3 mm, *v* = 80 mm/s, *t* = 0.2 mm, *I* = 30%, *θ_p_* = 90 °C, *θ_e_* = 245 °C, according to the determined empirical mathematical models, PETG links printed in a horizontal position will break for a force of 300.8 N, while links printed in a vertical position will break for force values of 35.8 N.

The chemical compositions of the four polymer materials from which the chain links were fabricated by 3D printing were different. They involved using different input factor values in the 3D printing process. To compare the results obtained in the case of using these different polymer materials, the chain links need to be printed under the conditions of using the same values of the input factors in the 3D printing process. In this way, it was possible that the used values of some of the input factors in the 3D printing process were not the most suitable for making links with high tensile strength. This aspect could be, for example, a cause of the relatively low value of the tensile strength determined in the case of gels made of polylactic acid. These printing conditions were the same for all the polymer materials from which the plates were made, being established in such a way that, as far as possible, they corresponded, to a greater or lesser extent, to the valid recommendations for the manufacture of parts from the respective polymer materials.

According to the determined empirical mathematical models, except for one material (PETG), the values of the maximum force *F* increase with the increase in the temperature *θ_p_* of the plate on which the chain links were generated. A possible explanation of this situation could be based on a better adhesion of successively deposited material layers when the plate temperature *θ_p_* increases.

It is also found that, except for the ABS-K material, increasing the velocity v leads to an increase in the maximum force *F*. It is possible that with increasing the velocity v, more favorable heat transfer conditions specific to the cooling of the specimen material are reached and, under these conditions, to achieve better adhesion of successively deposited layers of material, so at a higher value of the magnitude of the maximum force *F*.

According to the determined empirical mathematical models, a high influence on the magnitude of the maximum force *F* is exerted by the extrusion temperature *θ_e_*. In all empirical mathematical models, it was found that the exponents attached to the temperature in the extrusion nozzle have large and positive values. This means an increase in the extrusion temperature *θ_e_* will significantly increase the magnitude of the maximum force *F*. The findings can be explained by the fact that when the extrusion temperature *θ_e_* is increased, there is an increase in the fluidity of the material to be deposited and, as such, a better connection between layers of material added successively.

A validation test of the determined empirical mathematical models was possible by conducting experimental tests for other combinations of the values of some input factors in the 3D printing process. Some such results led to maximum force values close to values determined using empirical mathematical models.

Some of the results determined in the manner previously described were as follows:For an ABS chain link, where the variables took the values *d* = 4 mm, *v* = 80 mm/s, *t* = 0.2 mm, *i* = 60%, *θ_p_* = 90 °C, *θ_e_* = 265°, *p* = 2 (printing in horizontal position), the maximum force *F* values were 460.84 N for the experimental test and 469 N when the empirical mathematical model was used. The difference between the two values is 1.9%.For a chain link made of ABS-K, where the variables took the values *d* = 4 mm, *v* = 80 mm/s, *t* = 0.2 mm, *i* = 30%, *θ_p_* = 90 °C, *θ_e_* = 245°, *p* = 2 (printing in horizontal position), the values of the maximum force *F* were 192.66 N for the experimental test and 284.21 N when the empirical mathematical model was used. The difference between the two values is 32.2%.For a chain link made of PETG, where the variables took the values *d* = 4 mm, *v* = 80 mm/s, *t* = 0.2 mm, *i* = 30%, *θ_p_* = 90 °C, *θ_e_* = 245°, *p* = 2 (printing in horizontal position), the maximum force *F* values were 564.53 N for the experimental test and 556.36 N when the empirical mathematical model was used. The difference between the two values is 1.46%.

Even larger differences were found for other sets of input factor values in the case of additional experimental tests performed. These differences could be explained by a rather large dispersion of the experimental results due, for example, to the variation and other input factors in the 3D printing process and whose values were not followed during the experimental trials. Large differences between the experimental results and those obtained using the empirical mathematical model were also observed in the case of chain links made of PLA. Such differences could also be generated by the fact that the values of some of the input factors in the 3D printing process were not suitable for the PLA material.

The highest value of the maximum force *F* was recorded in the case of the chain link made of PETG material. It should be noted that the value of the maximum force *F* is almost 7-fold higher when the chain link is made in a horizontal position compared to that made in a vertical position. A possible explanation for this was mentioned earlier. PETG polymer material also has the highest tensile strength, as seen in Table 1.

## 4. Conclusions

The problem of replacing parts made of metallic materials with parts made of plastic materials manufactured by 3D printing can also be formulated in the case of chain links. For reduced mechanical stress, link chains made of polymeric materials have a lower weight and better corrosion resistance than chains made of metallic materials. It should be noted that such chains made of polymer materials could be manufactured directly by 3D printing without assembling separately manufactured chain links. The consultation of specialized literature highlighted research concerns in such a direction. The research presented in this article aimed to highlight the influence exerted by some input factors in the 3D printing process of chain links from four different materials on the tensile behavior of the respective chain links. The four materials from which the chain links were made were acrylonitrile butadiene styrene (ABS), the composite material acrylonitrile butadiene styrene-Kevlar (ABS-K), polylactic acid (PLA) and polyethylene terephthalate glycol (PETG). The experimental tests were conducted according to the requirements of a fractional factorial experiment with seven input factors at two levels of variation. The chain link diameter in a cross-section, printing speed, layer thickness, infill density, the temperature of the printer table, extrusion temperature, and test sample placement during 3D printing were considered input factors. Through the mathematical processing of the experimental results, empirical mathematical power-type function models were determined. These empirical mathematical models provide information on the direction of variation and the intensity of the influence exerted by the input factors considered on the maximum force at which damage to the chain links occurs. The analysis of the experimental results highlighted the fact that the same directions of action of the input factors were recorded only in the case of the diameter of the chain link rod and the way of positioning the sample during printing. As expected, printing the chain link horizontally ensured superior mechanical strength due to the stress development along the deposited layers, thus characterized by a certain continuity. The results of the experimental research showed that chain links printed in a horizontal position are almost 9-fold stronger than chain links printed in a vertical position. Under the same test conditions, according to the determined empirical mathematical models, PETG links printed horizontally will break for a force of 300.8 N. In comparison, links printed in a vertical position will break for force values of 35.8 N. Polyethylene terephthalate glycol (PETG) links have proven the highest mechanical resistance among the materials used. Under the same test conditions, the PETG links broke for a force of 40.6 N, while the polylactic acid rings broke for values of 4.70 N. In the future, it is intended to continue the research by considering other materials for the chain links and identifying more appropriate mathematical and empirical models to illustrate the tensile behavior of some plastic materials incorporated in the chain links. Another research direction could be to directly manufacture complete chains with polymer links assembled by the 3D printing process itself, eliminating the need for an additional operation of assembling separately manufactured links.

## Figures and Tables

**Figure 1 polymers-15-03178-f001:**
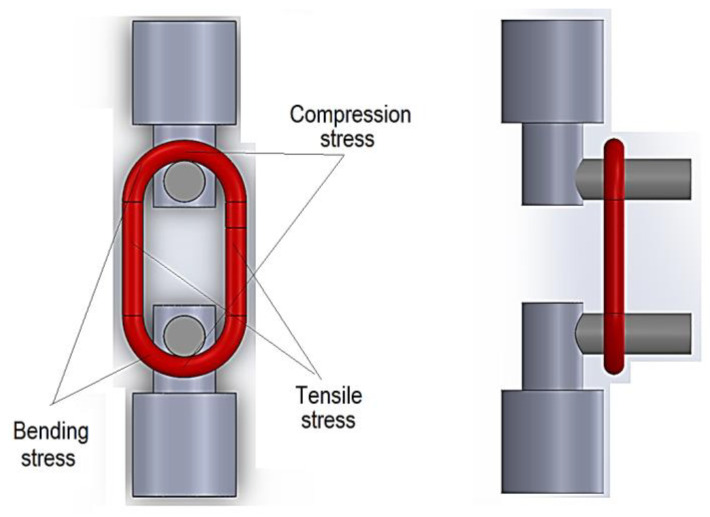
Way to clamp the chain link on the tensile test machine, using two bolts.

**Figure 2 polymers-15-03178-f002:**
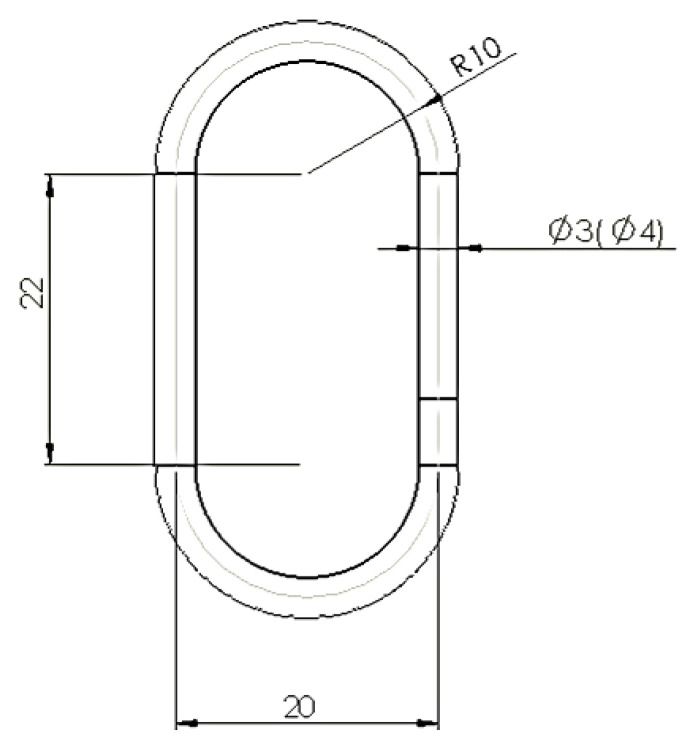
Dimensions of the chain link proposed to be tested.

**Figure 3 polymers-15-03178-f003:**
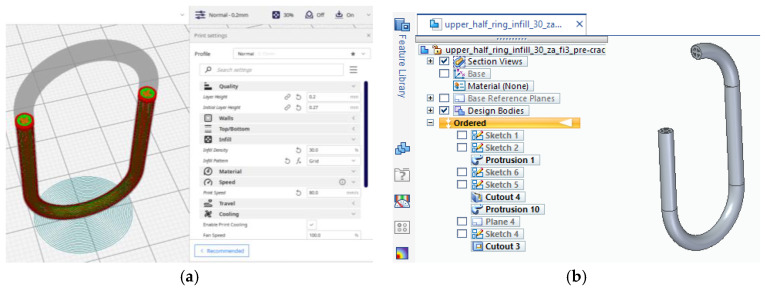
Models of the chain link: (**a**)—intermediate step from the preview of the 3D printing process; (**b**)—section view of the 3D model with a 30% infill pattern.

**Figure 4 polymers-15-03178-f004:**
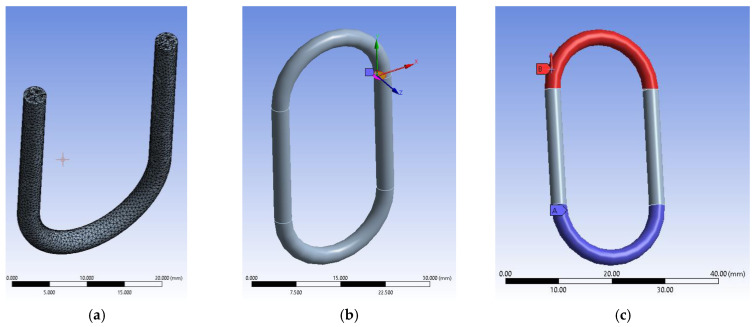
FEM setup: (**a**)—section view of the meshed model with a 30% infill patterns; (**b**)—view of the pre-meshed crack setup with its coordinate systems; (**c**)—setup conditions inside analyses settings.

**Figure 5 polymers-15-03178-f005:**
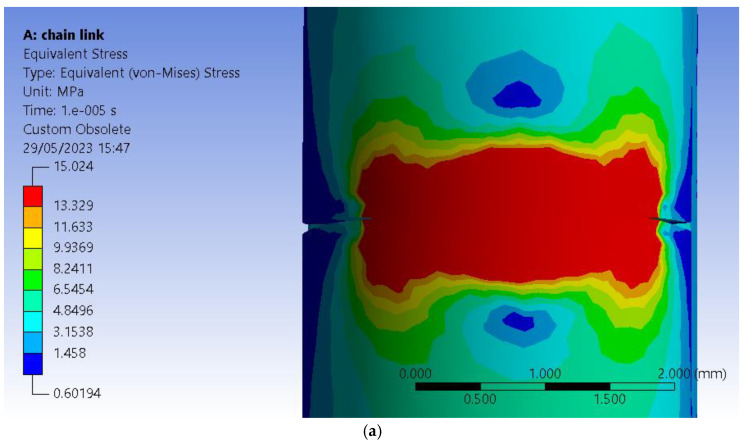
Results of FEM and experimental test: (**a**)—side view of the distribution of equivalent (von Mises) stress type of result; (**b**)—front view of the distribution of equivalent elastic strain type of result; (**c**)—front view of the real-life cracked sample 1 made of ABS.

**Figure 6 polymers-15-03178-f006:**
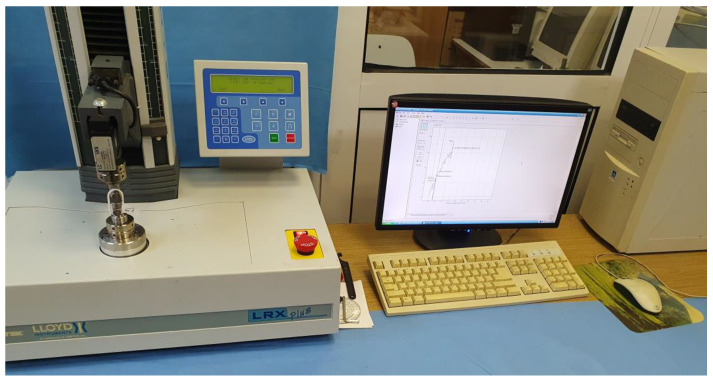
Components of equipment used for tensile testing of chain links.

**Figure 7 polymers-15-03178-f007:**
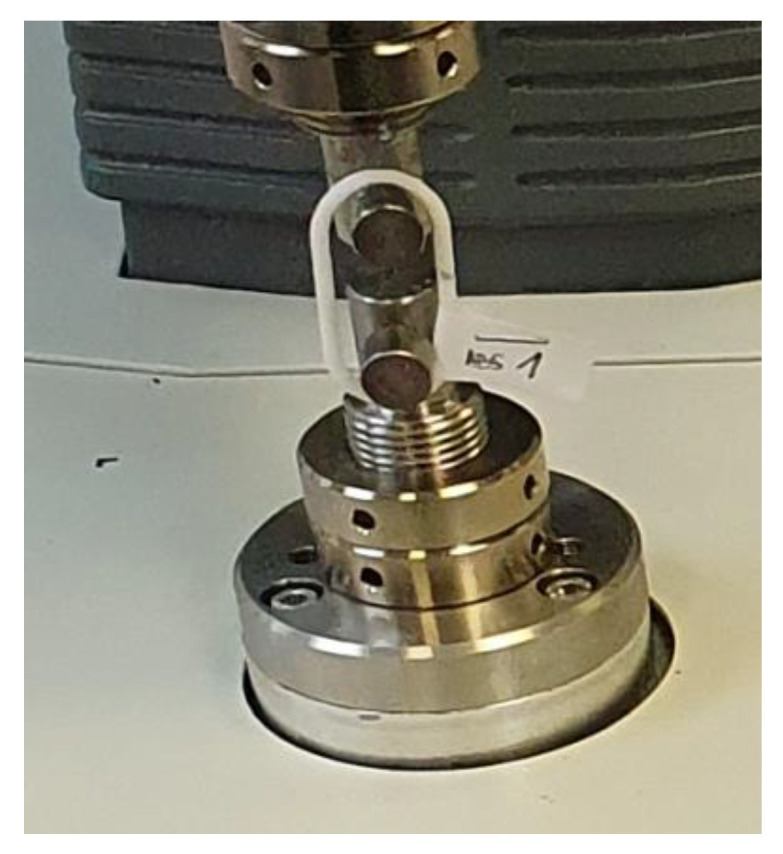
Clamping the chain link on the tensile testing machine.

**Figure 8 polymers-15-03178-f008:**
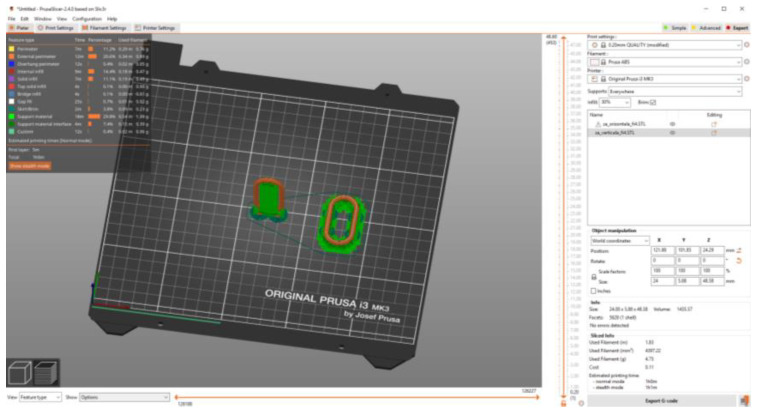
Screenshot showing how to place the printed chain link vertically or horizontally.

**Figure 9 polymers-15-03178-f009:**
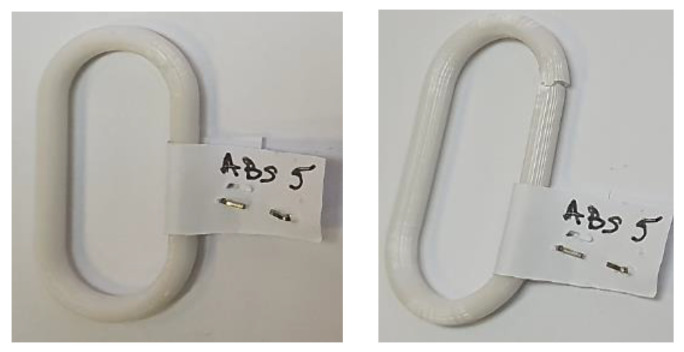
The appearance of a chain link made of ABS polymer material before (**left**) and after (**right**) breaking, respectively.

**Figure 10 polymers-15-03178-f010:**
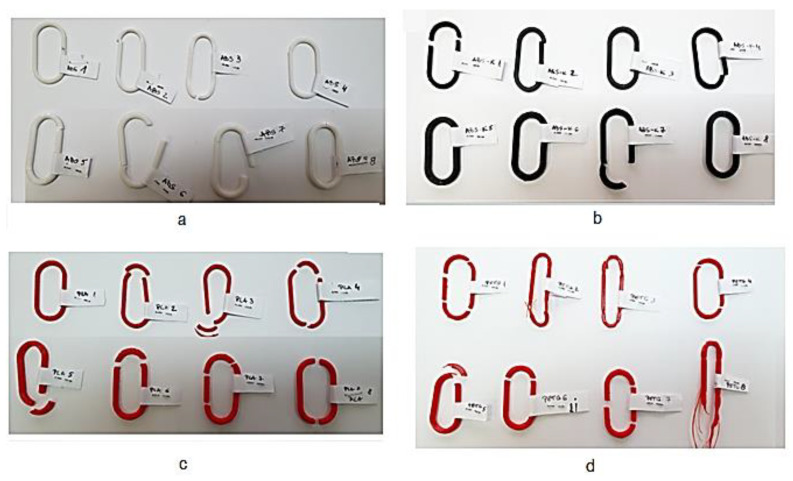
Images of the 32 links manufactured by 3D printing from 4 different polymer materials, after breaking them by the tensile test ((**a**)—ABS links; (**b**)—ABS-K links; (**c**)—PLA links; (**d**)—PETG links).

**Figure 11 polymers-15-03178-f011:**
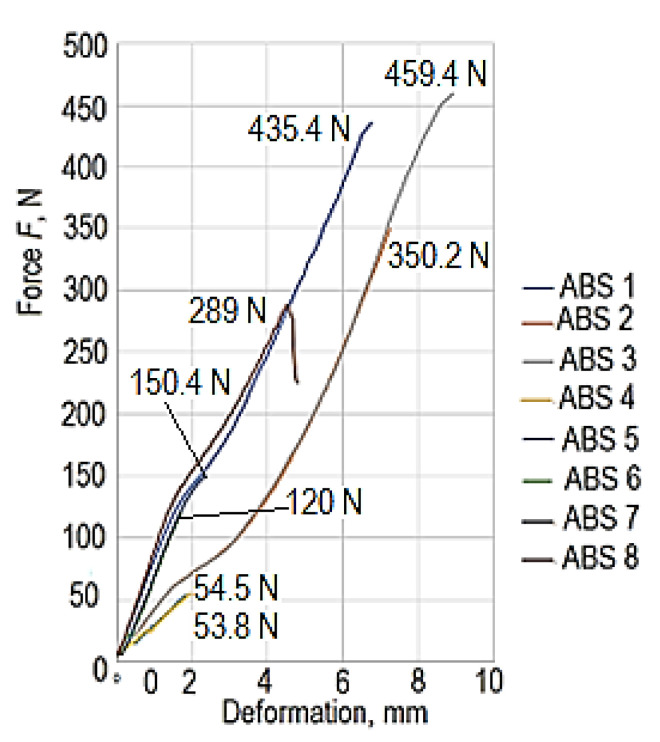
Examples of force–deformation curves in the case of links made of ABS polymer material, corresponding to the 8 experimental tests.

**Figure 12 polymers-15-03178-f012:**
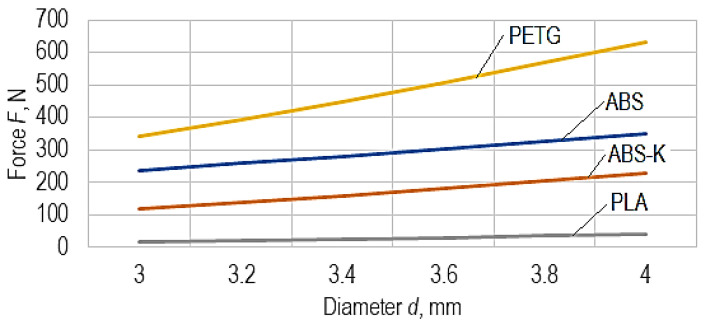
The influence of the diameter *d* on the magnitude of the maximum force *F* that determines the damage of a chain link made of different materials (*v* = 80 mm/min, *t* = 0.2 mm, *I* = 30%, *θ_p_* = 90°, *θ_e_* = 245°, *p* = 2, corresponding to manufacturing in horizontal position).

**Figure 13 polymers-15-03178-f013:**
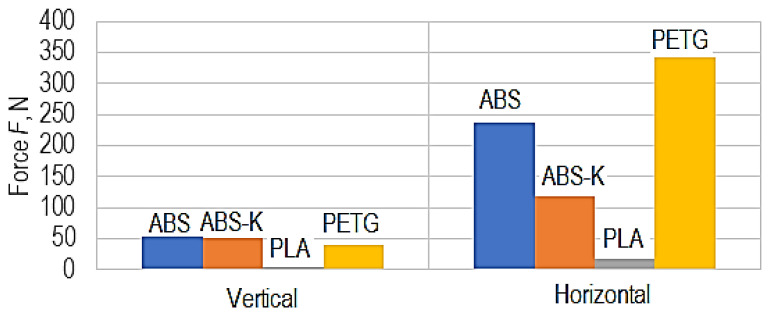
Differences between the maximum force values for the four polymeric materials considered in the case of manufacture in vertical and horizontal positions, respectively (d = 3 mm, v = 80 mm/min, *t* = 0.2 mm, *i* = 30%, *θ_p_* = 90 °C, *θ_e_* = 245 °C).

**Table 1 polymers-15-03178-t001:** Some properties of the materials used in the case of tensile-tested beams.

Properties	ABS	ABS-Kevlar	PLA	PETG
Tensile modulus [GPa]	1.6815 (ISO 527)	1.775 (ISO 527)	2.3 (ASTM D638—Type V)	1.6 ± 0.1 (ISO 527-1—vertical XZ)
Tensile strength [GPa]	0.0436 (ISO 527—at yield)	0.0311 (ISO 527—at yield)	0.0359 (ASTM D638—Type V—at yield)	0.05 (ISO 527-1—vertical XZ)
Elongation	3.5 % (ISO 527—at yield)	2.3 % (ISO 527—tensile strength)	2 % (ASTM D638—Type V—at yield)	5.1 % (ISO 527-1—at yield)
Hardness	97 (Shore A)	65.2 (Shore D)	95 (Shore D)	74 (Shore D)

## Data Availability

The data presented in this study are available on request from the corresponding author. The data are not publicly available due to privacy.

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
