# Peer review of "Tensile Behavior of Chain Links Made of Polymeric Materials Manufactured by 3D Printing"

_polymers, 2023, doi:10.3390/polym15153178_

Round 1
Reviewer 1 Report
This paper presents numerical and experimental investigations into the tensile behaviours of 3D-printed polymer chain links. However, I must reject the paper due to flaws in the significance of the study, methodology, and discussion aspects.
Firstly, the study focuses on 3D-printed polymeric chain links because there is no related research on this topic. While I acknowledge this as a research gap, the authors did not consider why people did not previously care about it. The reason is that FDM polymers cannot provide sufficient strength for real-world applications, even compared to injection moulding polymers. This is a fundamental flaw of FDM structures. Thus, for the first sentence in the abstract, 'Replacing relatively heavy steel chain links with lighter but sufficiently resistant polymer material chain links may be necessary for some practical situations,' I highly doubt its practicality. For example, the authors also mentioned the use of metal chain links in ship mooring; it is difficult to imagine replacing these chains with 3D-printed polymers. The low strength of FDM structures significantly limits their applications in real environments. Therefore, this study is relatively meaningless. I hope the authors can provide some real scenarios/cases/examples where heavy steel chain links could be replaced with 3D-printed ones.
Secondly, due to the large number of repeated chain links, injection moulding would be a better manufacturing method compared to FDM. Injection moulding is more suitable for the mass production of chain rings with the same simple geometries, while FDM is better suited for customized, complex structures, and prototyping.
For the numerical works, the authors directly assigned the ABS material properties from the software's library. However, the actual modulus of FDM ABS is definitely lower than the nominated value in the software due to the nature of FDM. Therefore, the approach described in the text is not acceptable, as it causes the structural performance to be overestimated. The correct method would be to measure the actual modulus through multiple tests and quote the measured values. However, due to the lack of manufacturing accuracy of FDM, I can foresee a huge difference in these moduli from repeated tests, making it very tricky to obtain robust and accurate numerical results.
The research also applied weird boundary conditions during the simulation. The authors only investigated the straight bar part of the link and assumed that the initial crack was there. However, in reality, the chain links were point-contacted at the semi-circular part. Therefore, the stress concentration location must not be at the straight bar. The assumption of the initial crack location, boundary conditions, and load must be redesigned.
Regarding the experimental works, the research only designed two levels for each variable. This is not enough for a comprehensive assessment, but the paper even proposed some mathematical correlation equations based on these poor results. Although the authors said they applied mathematical power-type function models, Equation (1-4) could easily be transferred to linear regression models. However, two levels of variables cannot provide the correct change trend of the force.
The authors also did not mention how many repeat tests were conducted for each printing combination to ensure the repeatability of the results. Additionally, I did not see reasonable justifications for the extremely low force of PLA.
Author Response
The authors of the reviewed paper wish to express their gratitude for the efforts of the reviewers invested in the analysis of the proposed paper and for the useful observations and suggestions to improve the quality of the paper.
All the changes were highlighted in the manuscript of the paper by using the color green.

Reviewer 2 Report
1. author should highlight the significant advantages of this polymeric chain link compared with traditional chain link
2. results should be included in the abstract and conclusion.
3. results on the stress-strain diagram should necessarily be included as it implies the title of this study
4. images of the sample should be shown
Author Response

(The authors gave the same response as above.)

Reviewer 3 Report
In this paper, authors have used 3D printing process to replace steel chain links with lighter materials such as polymers. They have applied tensile test to be able to characterize the mechanical characteristic of the fabricated parts in order to determine the role of different process variables. They have then performed a discussion on the characteristic of the applied material to be able to propose for the desired application.
The paper is interesting and I propose it for publication after addressing the following highlights:
1. Introduction: there should be more discussion on the literature and the performed discussion is not sufficient. I propose to improve the introduction by including more related works and you can also consider this reference that is related to the mechanical properties of the fabricated materials by 3D printing: https://doi.org/10.3390/ma15248722
2. Materials and methods: what is the reference for the applied geometry ? it is not clear and it should be mentioned based on which reference you have used this geometry.
3. Results and discussion: I propose to include more discussion on the obtained results. It seems just like a report of some results from a work and it is required to be compared with some similar works.
4. Conclusion and abstract: I propose to include more quantitative results in the abstract as you have already included in the conclusion.
Best,
Author Response

(The authors gave the same response as above.)

Round 2
Reviewer 1 Report
Please find the attached file.

Author Response
Once again, the authors wish to express their thanks for the reviewers' attention to the article's content, and for comments and suggestions aimed at improving the content of the article.
The new changes introduced in the article have been highlighted using blue.
REVIEWER 1
Authors’answers to the reviewer comments. In principle, the reviewer is right in some of his comments. For some of the comments, we are reluctant to accept them.
Thus, the reviewer considers that carrying out experimental tests only at two levels is not acceptable, since the number of variation levels (two) is too small.
First of all, we mention that the HYPOTHESIS (and not the CERTAINTY) was formulated that for the assumed variation intervals of the independent variables (of the input factors in the studied process), we will have a monotonous variation of the output factor (of the dependent variable). In the text of the article (variant after the first round), in lines 349-352, there is the wording ”It will be assumed, as such, that in the situations presented in this paper and for the variation intervals considered, the seven input factors do not determine the occurrence of maxima or minima of the value of the force F, and they lead to a monotonous variation of the output parameter”. It is entirely the right of the reviewer to accept or not such a HYPOTHESIS.
Second, in lines 347-349, it is stated that ”It is necessary to note that power-type functions are preferred when it is considered that there is a monotonous variation of the pursued output parameter to the variation of the input factors in the investigated process".
This means that the use of power-type functions was PREFERRED since, as mentioned in line 345-347, such functions can often be found in manufacturing processes (we mentioned in the article that such relations were used to characterize ”the tool life, cutting force components, roughness parameters of the machined surfaces, etc.” (lines 346-347) and bibliographical references can be identified to justify the previous statement.
It should be noted that other scientific articles have also used two levels of variation and factorial experiments of the L8-27 type (with seven variables having values at two levels of variation). A quick web search has so far identified the following 7 articles in which results of factorial L8 experiments were used (27):
- Kokcam, A.H.; Uygun, Ö.; Taskin, M.F., Demir, H.I., Demir, Z. Modelling Porosity Permeability of Ceramic Tiles using Fuzzy Taguchi Method, Open Chem. 2018, 16, 1111–1114.
- Constructing Orthogonal Arrays. Available: https://ocw.mit.edu/courses/16-881-robust-system-design-summer-1998/dc71137bc61a68bd65d8416928727c07_l8_orth_arrays.pdf, accessed: 1.07.2023, p. 5
- Agarwal, S.;Tyagi, I.; Gupta, V.K., Jafari, M., Edrissi, M., Javadian, H. Taguchi L8 (27) orthogonal array design method for the optimization of synthesis conditions of manganese phosphate (Mn3(PO4)2) nanoparticles using water-in-oil microemulsion method. Journal of Molecular Liquids, 219, 2016, 1131-1136, https://doi.org/10.1016/j.molliq.2016.04.022
- Sukthomya, W., Tannock, J.D.T.Taguchi experimental design for manufacturing process optimization using historical data and a neural network process model. International Journal of Quality & Reliability Management, 2005, 22, 5, 485-502. Available: https://www.emerald.com/insight/content/doi/10.1108/02656710510598393/full/html, accessed: 1.07.2023
- Deprez, P., Melian, C.-F., Breaban, F., Coutouly, J.-F.. Glass Marking with CO2Laser: Experimental Study of the Interaction Laser-Material. Journal of Surface Engineered Materials and Advanced Technology, 2012, 2, 1, 16996, DOI:10.4236/jsemat.2012.21006
- Kaushik, G., Thakur, I.S. Isolation of fungi and optimization of process parameters for decolorization of distillery mill effluent. World J Microbiol Biotechnol, 2009 25, 6, 157-163 DOI 10.1007/s11274-009-9970-0
- Sorgdrager, A., Wang, R.-J., Grobler, A.. Taguchi method in electrical machine design. SOUTH African Institute of Electrical Engineers, 108, 4, 2017, 150-164, DOI: 10.23919/SAIEE.2017.8531928
In the following 6 papers, there are also recommendations for using factorial experiments of type L8 (27), so with seven independent variables at two levels of variation:
- Robust Design: Statistical Analysis for Taguchi Methods. Available: https://ieda.ust.hk/dfaculty/ajay/courses/ielm317/lecs/robust/robustdesign2.pdf, accessed: 1.07.2023, Tabel 2, p. 2.
2. University of Yorks. Department of Mathematics Orthogonal Arrays (Taguchi Designs). Available: https://www.york.ac.uk/depts/maths/tables/l8.gif, accessed: 1.07.2023
- Kumar, V. Taguchi Method Experimental Design Technique. International Conference on Science, Technology and Management (ICSTM-2020), pp. 22-29, available: http://proceeding.conferenceworld.in/ICSTM2020/4Nrdnbz646wxN715.pdf, accessed: 1.07.2023
- Kai-Tai Fang, Min-Qian Liu, Hong Qin, Yong-Dao Zhou. Theory and Application of Uniform Experimental Designs. Springer, 2018. Available: http://ndl.ethernet.edu.et/bitstream/123456789/75946/1/2018_Book_TheoryAndApplicationOfUniformE.pdf, accessed: 1.07.2023, p. 18, Table 1.3
5. Bolboacă, S.D., Jäntschi, L. Design of Experiments: Useful Orthogonal Arrays for Number of Experiments from 4 to 16. Entropy 2007, 9, 4, 198-232, available: https://doi.org/10.3390/e9040198, accessed: 1.07.2023
6. O’Reilly. Appendix C. Some Useful Orthogonal Arrays. Available: https://www.oreilly.com/library/view/competing-with-high/9781118416495/bapp03.xhtml, accessed: 1.07.2023
It may be noted that among the 18 arrays proposed by Taguchi, 12 arrays consider variables with only two levels of variation, and only 6 arrays do not consider variables with two levels of variation. In many of the previously mentioned papers (but also in other papers, since in the cited papers only factorial experiments of the L8 type are used (27), this is only one of the 12 Taguchi tables that consider two levels of variation of a factor of input!), the authors adopted the assumption of using only two levels of variation and elaborated the commonly known plots of mean responses for the independent variables on only the two levels of variation.
The reviewer recommended using an L18 Taguchi experiment. Such an experiment presupposes (according to Pillet, M. Introduction aux plans d’expériences par la méthode Taguchi. Paris: Les Éditions d’Organisation, 1992, Deuxième edition: 1994, etc.) using 8 independent variables (8 input factors), namely one independent variable with a two-level variance and 7 independent variables with a 3-level variance. According to the reviewer, this means that, ALSO IN THIS CASE, one could object that there might be too few levels of variation for the single independent variable with only two levels of variation. In addition, the number of experimental tests required to use an L18-type Taguchi experiment increases more than twice. In addition, there is the problem of including a new independent variable (the Taguchi L8 experiment considers 7 independent variables, while the Taguchi L18 experiment requires the consideration of 8 independent variables). We appreciate that in the relatively short time frame for formulating the answers to the reviewers' comments, it would have been quite difficult for us to carry out 18 experimental trials and to provide the mathematical processing and interpretation of the results of the 18 experimental trials.
We will also add that, according to our knowledge, most software for identifying empirical mathematical models (perhaps even all software with such a destination!) are based on the method of least squares, and they also provide the possibility of determining empirical mathematical models values corresponding to non-monotonic functions (so they have maxima or minima). Including within the software used by us, we have shown that through the mathematical processing of the results of the 8 experimental tests, non-monotonic functions can also be identified (in lines 331-332, second-degree polynomial and hyperbolic-type functions were thus mentioned and THEY ARE NOT MONOTONE FUNCTIONS).
The internal working mode of the software based on the method of least squares takes into account all 8 experimental results obtained as if a diagram should be elaborated on 8 points corresponding to the 8 experimental results (and not only those two results corresponding to the two levels of an independent variable with two levels of variation). This way of working provides the conditions for identifying some empirical mathematical models that are not monotonic functions. However, we have previously outlined the reasons for preferring a power function.
In the reviewer's comments after the first review stage, reviewer 1 shows that "Your models (Equations 1 to 4) are monotonic and cannot model the possible maxima or minima of the output parameter. Therefore, I am afraid I have to disagree with your models. In my first-round comment, I never suggested using linear regression. Quite the opposite, I pointed out that the manuscript's models (Equations 1 to 4) are equivalent to linear regression”. We apologize, but from the wording (included in the first review) "Equation (1-4) could easily be transferred to linear regression models.", we thought that the reviewer was expressing his option to use a linear regression model.
Regarding the linearization of a function by logarithmization, we will mention the fact that most software (perhaps even all software based on the use of the least squares method!) for processing experimental results for the identification of empirical mathematical models (of regression functions), use, in their working program, a linearization by logarithmization, to arrive at systems of equations of the first degree, easier to solve by known methods and thus be able to determine those constants (coefficients and exponents) that will be included in the proposed empirical mathematical models.

Reviewer 2 Report
All the results should be shown precisely and mentioned in the abstract and conclusion. Put the exact value.
page 2 line 88: Please elaborate more
page 6 line 231: please provide the proof
Please rearrange the figures accordingly
Author Response
Once again, the authors wish to express their thanks for the reviewers' attention to the article's content, and for comments and suggestions aimed at improving the content of the article.
The new changes introduced in the article have been highlighted using blue.
REVIEWER 2
Reviewer's comment no. 1. All the results should be shown precisely and mentioned in the abstract and conclusion. Put the exact value.
Authors response to the reviewer's comment. The authors considered that the reviewer was right. Texts consistent with the author's suggestions were mentioned in the Abstract and in the conclusions section.
Reviewer's comment no. 2. page 2 line 88: Please elaborate more
Authors response to the reviewer's comment. Appreciating that the reviewer is right, the following text was added to the article:
“Woodman produced a report outlining the possibilities of using 3D printing processes to create objects that could improve the living conditions of pets or laboratory animals. He thus analyzed the situation of the use of chain links made of plastic materials, appreciating that the plastic material of the links must be hard enough so that the animals do not swallow the particles detached from the links as a result of use and harm their health”.
Reviewer's comment no. 3. page 6 line 231: please provide the proof
Authors response to the reviewer's comment.
The manuscript has a related image shown in Figure 5c. However, we provide several images from different angles for ABS sample 1. Our FEM model showed a similar behavior when the pre-designed crack had a similar propagation direction, as we can see in the following images.
Reviewer's comment no. 4. Please rearrange the figures accordingly
Authors response to the reviewer's comment. The authors have considered the reviewer's comment, but the assigned editor will decide the final arrangement of the figures.

Round 3
Reviewer 1 Report
Dear authors,
Thank you again for your outstanding revisions. I have just one small suggestion now.
It would be better to cite some references as justifications supporting your monotonic assumption in Lines 357-360. Because in my view, your hypothesis was proposed for the purpose of simplifying experiments. It was not proposed based on some previous research outcomes or fundamental knowledge.
This lack of theoretical justification should be addressed. E.g. Find some references introducing how input factors affect the microstructures and then affect the ultimate tensile strength. They may provide some clues about the monotonic trend of the force.
Author Response
Thank you for suggestions and we tried to cover the info requested.